# Monitoring and Forecasting the Impact of the 2018 Summer Heatwave on Vegetation

**Clément Albergel [1,*]**, **Emanuel Dutra [2]**, **Bertrand Bonan [1]**, **Yongjun Zheng [1]**,
**Simon Munier [1]**, **Gianpaolo Balsamo [3]**, **Patricia de Rosnay [3]**, **Joaquin Muñoz-Sabater [3]**
and **Jean-Christophe Calvet [1]**

1   CNRM—Université de Toulouse, Météo-France, CNRS, 31057 Toulouse, France;
    bertrand.bonan@meteo.fr (B.B.); yongjun.zheng@meteo.fr (Y.Z.); simon.munier@meteo.fr (S.M.);
    jean-christophe.calvet@meteo.fr (J.-C.C.)
2   Instituto Dom Luiz, IDL, Faculty of Sciences, University of Lisbon, 1749-016 Lisbon, Portugal;
    endutra@fc.ul.pt
3   European Centre for Medium-range Weather Forecasts (ECMWF), Reading RG2 9AX, UK;
    Gianpaolo.Balsamo@ecmwf.int (G.B.); patricia.rosnay@ecmwf.int (P.d.R.);
    Joaquin.Munoz@ecmwf.int (J.M.-S.)
*   Correspondence: clement.albergel@meteo.fr

**Abstract:** This study aims to assess the potential of the LDAS-Monde platform, a land data assimilation system developed by Météo-France, to monitor the impact on vegetation state of the 2018 summer heatwave over Western Europe. The LDAS-Monde is driven by ECMWF's (i) ERA5 reanalysis, and (ii) the Integrated Forecasting System High Resolution operational analysis (IFS-HRES), used in conjunction with the assimilation of Copernicus Global Land Service (CGLS) satellite-derived products, namely the Surface Soil Moisture (SSM) and the Leaf Area Index (LAI). The study of long time series of satellite derived CGLS LAI (2000–2018) and SSM (2008–2018) highlights marked negative anomalies for July 2018 affecting large areas of northwestern Europe and reflects the impact of the heatwave. Such large anomalies spreading over a large part of the domain of interest have never been observed in the LAI product over this 19-year period. LDAS-Monde land surface reanalyses were produced at spatial resolutions of $0.25° \times 0.25°$ (January 2008 to October 2018) and $0.10° \times 0.10°$ (April 2016 to December 2018). Both configurations of LDAS-Monde forced by either ERA5 or HRES capture well the vegetation state in general and for this specific event, with HRES configuration exhibiting better monitoring skills than ERA5 configuration. The consistency of ERA5- and IFS HRES-driven simulations over the common period (April 2016 to October 2018) allowed to disentangle and appreciate the origin of improvements observed between the ERA5 and HRES. Another experiment, down-scaling ERA5 to HRES spatial resolutions, was performed. Results suggest that land surface spatial resolution is key (e.g., associated to a better representation of the land cover, topography) and using HRES forcing still enhances the skill. While there are advantages in using HRES, there is added value in down-scaling ERA5, which can provide consistent, long term, high resolution land reanalysis. If the improvement from LDAS-Monde analysis on control variables (soil moisture from layers 2 to 8 of the model representing the first meter of soil and LAI) from the assimilation of SSM and LAI was expected, other model variables benefit from the assimilation through biophysical processes and feedback in the model. Finally, we also found added value of initializing 8-day land surface HRES driven forecasts from LDAS-Monde analysis when compared with model-only initial conditions.

**Keywords:** land surface modeling; data assimilation; leaf area index; surface soil moisture; summer 2018 heatwave

## 1. Introduction

Land surface conditions are critical in the global weather and climate system. Accurate characterization and simulation of hydrological and biophysical variables at the land surface represent a significant challenge given large spatial heterogeneity and human modifications of the land surface. In particular, observing and simulating the response and feedback of land surface conditions to extreme events is crucial in our ability to manage adaptation to climate change impacts. Land Surface Model's (LSM's) role has evolved over the years, from the primary goal of providing boundary conditions to atmospheric models to being used as monitoring and forecasting tools for estimating land surface conditions [1–4].

Modeling of terrestrial variables can be improved through the dynamical integration of observations [5–7] and there is a growing emphasis on constraining LSM estimates with observational inputs as well as coupling them with other models of the Earth system [1,8–10]. Enhanced estimates of land surface conditions are also recognized to improve forecasts of weather patterns, sub-seasonal temperatures and precipitations, agricultural productivity, seasonal streamflow, floods and droughts, as well as the carbon cycle [11–16]. Remote sensing observations are particularly useful in this context as they are now unrestrictedly available at a global scale with high spatial resolution and with long-term records. Many satellite-derived products relevant to the hydrological (e.g., soil moisture, snow depth/cover, terrestrial water storage), vegetation (e.g., leaf area index, biomass), and energy (e.g., land surface temperature, albedo) cycles are readily available [17].

Data assimilation techniques allow to spatially and temporally integrate the observed information into LSMs in a consistent way [5,18]. We refer to Land Data Assimilation Systems (LDASs) as the framework where LSMs are driven by and/or ingest such observations generating enhanced estimates of the land surface variables (LSVs) [10]. Several LDASs now exist from point to regional scale, amongst them are the Global Land Data Assimilation System (GLDAS, [19]), the Carbon Cycle Data Assimilation System (CCDAS, [20]), the Coupled Land Vegetation LDAS (CLVLDAS, [21,22]) and more recently the U.S. National Climate Assessment LDAS (NCA-LDAS, [10]) as well as LDAS-Monde [7,18] to name a few. These LDASs either optimize process parameters (e.g., CCDAS), state variables (e.g., GLDAS, NCA-LDAS, LDAS-Monde) or both (e.g., CLVLDAS). Assimilated Earth Observations (EOs) generally include satellite retrieval of surface soil moisture [5,8,23–25], snow depth [26–29] and snow cover [9,27,30,31], vegetation [7,18,32–35], as well as terrestrial water storage [36–38]. Few studies have included multiple remote sensing measurements. For instance, Kumar et al. [10] assimilates various remote sensing measurements of the terrestrial water cycle within the NCA-LDAS over the USA while LDAS-Monde [7,18] considers the joint assimilation of vegetation (Leaf Area Index, LAI) and surface soil moisture (SSM) measurements. LDAS-Monde is a sequential land data assimilation system with global capacity. It has been evaluated over various domains at various spatial resolutions including France at 8 km scale [33,39] forced by the SAFRAN reanalysis of Météo-France (Système d'Analyse Fournissant des Renseignements Atmosphériques à la Neige [40,41], Europe at 0.5° × 0.5° [18,35] forced by ERA-Interim atmospheric reanalysis from the European Centre For Medium Range Weather Forecast (ECMWF) [42]; North America [7] and Burkina-Faso in western Africa at 0.25° × 0.25° [43] forced by ERA5 atmospheric reanalysis [44]. In those studies, analysis impact was successfully evaluated using several datasets such as (i) in situ measurements of soil moisture (ii) agricultural statistics, (iii) river discharge, (iv) independent flux estimates related to vegetation dynamics (evapotranspiration, Sun-Induced Fluorescence (SIF) and Gross Primary Productivity (GPP)). Albergel et al. [7] highlighted LDAS-Monde capacity to better characterize agricultural droughts (spatial area and intensity) than an open-loop counterpart (i.e. model without any assimilation of satellite derived measurements) over the continental United States of America. They found that LDAS-Monde can provide improved initial conditions to initialize forecast and that its impacts persist through time. In the above mentioned study, LDAS-Monde satellite-derived surface soil moisture dataset (ESA CCI SSM, [45–48]) along with satellite derived LAI (GEOV1,

http://land.copernicus.eu/global/ last access on June 2018), were jointly assimilated leading to a quarter degree spatial resolution reanalysis of the LSVs over 2010–2016.

Stemming from previous work [7], the present study investigates the capability of LDAS-Monde to represent the impact of the summer 2018 heatwave in Europe on vegetation. Spring and summer 2018 in Europe were marked by unusually hot weather that had led to record-breaking temperatures in many countries across Northern and Central Europe. According to ECMWF, near-surface air temperature anomaly in Europe in the period of April to August, calculated with respect to the 1981–2010 average for those months, was much larger in 2018 than in any previous year since 1979 [49]. According to the National Oceanic and Atmospheric Administration (NOAA) Europe had its second warmest July on record. It follows its second warmest June on record (behind 2003), its warmest May since continental records began in 1910, surpassing the previous record set in 2003: the whole summer 2018 was above climatology; Europe's warmest summer since continental records began in 1910 at +2.16 °C (Global Climate Report, https://www.ncdc.noaa.gov/sotc/global/ last access October 2018). Northern Hemisphere summer precipitation was generally weaker than normal across central Europe.

Such an event is likely to affect land surface conditions. In this study, satellite derived estimates of LAI and SSM as well as LDAS-Monde are used to monitor the impact of the heatwave on vegetation, focusing on July 2018. Firstly, we assess the heatwave impact on satellite derived LAI and SSM, using time-series over 2000 to 2018 and 2008 to 2018, respectively. Secondly, we evaluate the heatwave impact on the simulated LAI from LDAS-Monde forced by ECMWF ERA-5 reanalysis from January 2008 to October 2018 at $0.25° × 0.25°$ and by ECMWF Integrated Forecasting System (IFS) high resolution operational analysis (HRES) from April 2016 to December 2018 at $0.10° × 0.10°$. The use of both ERA5 and HRES to force LDAS-Monde enable to assess the impact of resolution versus system quality over a common one year period (2017) were ERA5 was downscaled to HRES spatial resolution. Another added value of using HRES consists in its forecast capacity, up to 10 days ahead. Forecast of LAI initialized by LDAS-Monde analysis with a leading time up to 8-days is then investigated in order to assess whether or not the heatwave impact on land surface conditions could have been anticipated. The remainder of this paper is organized as follows: Section 2 describes the LDAS-Monde system, the satellite derived estimates of LAI and SSM and the ECMWF analysis and reanalysis forcing, results are analyzed and discussed in Sections 3 and 4.

## 2. Material and Methods

This study assesses the ability of LDAS-Monde sequential assimilation of satellite derived surface SSM and LAI to represent the impact of the summer 2018 heatwave in Europe on vegetation. The following sections describe LDAS-Monde system as well as 2 other key elements of its setup: atmospheric forcing (LDAS-Monde being an offline system) and satellite derived observations.

### 2.1. LDAS-Monde

Within the SURFEX modeling platform of Météo-France (Surface Externalisée, [50], Version 8.1), the LDAS [32–34,39,51] developed in the research department of Météo-France, the CNRM (Centre National de Recherches Météorologiques) permits integrating satellite products into the ISBA (Interaction between Soil Biosphere and Atmosphere) LSM [52–55] using a data assimilation scheme. The LDAS was extended to a global scale (LDAS-Monde, [18]). At the same time, the coupling to hydrological models (ISBA-CTRIP for ISBA-CNRM, Total Runoff Integrating Pathways) was consolidated. A full description of the ISBA-CTRIP system is presented in [56]. The obtained land surface reanalyses from LDAS-Monde account for the synergies of various upstream products (e.g., model and satellite derived observations) and are able to provide an improved representation of LSVs, as well as statistics which can be used to monitor the quality of the assimilated observations (e.g., [7,18,35]). LDAS-Monde can also be used to calibrate model parameters (e.g., [57] for the soil maximum available water content within ISBA).

LDAS-Monde uses the $CO_2$-responsive [53–55], multi-layer soil [56–59], version of ISBA. The latter allows to solve the energy and water budgets at the surface level and describes the exchanges between the land surface and the atmosphere. Parameters in ISBA are defined for 12 generic land surface patches: nine plant functional types (namely: needle leaf trees, evergreen broadleaf trees, deciduous broadleaf trees, C3 crops, C4 crops, C4 irrigated crops, herbaceous, tropical herbaceous, and wetlands, C3 and C4 refer to different processes that plants use to fix carbon during the process of photosynthesis) as well as bare soil, rocks, and permanent snow and ice surfaces. They are derived from ECOCLIMAP-II, the land cover map used in SURFEX [60]. Atmospheric and climate conditions drive the dynamic evolution of the vegetation biomass and LAI through vegetation growth and mortality processes implemented in the form of a nitrogen dilution process (NIT option) [53,55,61]. Photosynthesis enables vegetation growth resulting from the CO2 net assimilation. During the growing phase, enhanced photosynthesis corresponds to a CO2 net assimilation, which results in vegetation growth from the LAI minimum threshold (1 $m^2m^{-2}$ for coniferous forest or 0.3 $m^2m^{-2}$ for other vegetation types). Vegetation phenology relies on photosynthesis-driven plant growth and mortality, and photosynthesis is related to the mesophyll conductance. More information on the $CO_2$-responsive version of ISBA can be found in [62,63] and by visiting the SURFEX website (www.umr-cnrm.fr/surfex, last access: February 2019). The multilayer diffusion scheme described in [58,59] drives transfers of water and heat through the soil. Finally, the Simplified Extended Kalman Filter Data Assimilation (DA) technique (SEKF, [18,32–34,39,51]) is the main technique available within LDAS-Monde. While ensemble based DA techniques are currently being tested and implemented [39,64], to date the LDAS-Monde SEKF is the most robust. It uses finite differences to compute the flow dependency between the assimilated observations (SSM and LAI) and the analyzed variables (soil moisture from soil layer 2 (1 cm to 4 cm) to layer 8 (80 cm to 100 cm), representing the first meter of soil and LAI, see Table 1). Further details of the analysis methodology can be found in [18,34]. While control variables are directly updated through their sensitivity to the observed variables, expressed by the SEKF Jacobians [18,65], other variables are indirectly modified by the analysis through biophysical processes and feedback in the model by updates of the control variables.

**Table 1.** Set up of the experiments used in this study.

| Experiments (Time Period) | Model | Domain and Spatial Resolution | Atm. Forcing | DA Method | Assimilated Observations | Observations Operators | Control Variables |
|---|---|---|---|---|---|---|---|
| LDAS-ERA5 * (01/2008-10/2018) | ISBA Multi-layer soil model $CO_2$-responsive version (Interactive vegetation) | Western Europe defined as longitudes from 10.5°W to 20.5°E, latitudes from 42°N to 59°N | ERA5 | SEKF | SSM (ASCAT) | Rescaled wg2 (Second layer of soil (1–4 cm)) | Layers of soil 2 to 8 (wg2 to wg8, 1–100 cm) |
| LDAS_HRES * (04/2016-12/2018) | | | IFS_HRES | | LAI (GEOV2) | LAI | LAI |
| ERA5_010 (2017) | | | ERA5 downscaled to 0.10° × 0.10° | 12-month model run | | | |
| LDAS_fc_d2 (2018) | | North Western Europe defined as longitudes from 5°W to 15°E, latitudes from 48°N to 55°N for the forecast experiments | IFS_HRES day 2 forecast | 12-month model run, every day a 2-day forecast initialized by an analysis is ran | | | |
| LDAS_fc_d8 (2018) | | | IFS_HRES day 8 forecast | 12-month model run, every day an 8-day forecast initialized by an analysis is ran | | | |

* Both LDAS_ERA5 and LDAS_HRES configurations consider an analysis (with data assimilation) and open-loop (model only, i.e., no data assimilation) experiments.

### 2.2. Satellite Derived Observations

Two satellite products from the Copernicus Global Land Service project are used in this study, the Surface Soil Moisture (SSM) and the Leaf Area Index (LAI) derived from SPOT-VGT (prior to 2014) and PROBA-V (from 2014 onward). The SSM product is derived from the Advanced Scatterometer (ASCAT), an active C-band microwave sensor on board the European MetOp polar-orbiting satellite (METOP-A&B). Information on soil moisture comes from ASCAT radar backscatter coefficients using a methodology developed at the Vienna University of Technology (TU-Wien) based on a change detection approach originally developed for the active microwave instrument flown on-board the European satellites ERS-1 and ERS-2 [66,67]. The recursive form on an exponential filter [68] is applied to the soil moisture product to estimate the Soil Wetness Index (SWI) using a timescale parameter, T, varying between 1 day and 100 days. T implicitly takes many physical parameters into account, it can be considered as a surrogate parameter for all the processes affecting the temporal dynamics of soil moisture, such as the thickness of the soil layer, soil hydraulic properties, evaporation, run-off and vertical gradient of soil properties (texture, density). T represents the time scale of soil moisture variation, in units of day. A high value of T describes a deeper soil layer (assuming that the soil diffusivity is constant) [69]. While giving a general rule on how to translate a given T-value to a certain soil depth is currently not possible since this depends strongly on the application and the soil composition of the area of interest, in this study, following work from [18,33] SWI-001 (i.e. T = 1 day, the smallest T value available) is used as a proxy for SSM [69]. The result for the top soil moisture content (<5 cm) is expressed as a degree of saturation and ranges between 0 (dry) and 100 (saturated). It is a global product at $0.1° \times 0.1°$ spatial resolution available daily from 2007. As in [7], pixels whose average altitude exceeds 1500 m above sea level as well as pixels with urban land cover fractions larger than 15% were discarded as those conditions may affect the retrieval of soil moisture from space.

SSM product has to be transformed into the model-equivalent surface soil moisture for data assimilation purposes and in order to address possible misspecification of physiographic model parameters (like field capacity and wilting point). As in [18] and [33] in LDAS-Monde, the linear rescaling approach described in [70] (using the first two moments of the cumulative distribution function, CDF) has been used in this study; it is a linear rescaling that enables a correction of the differences in the mean and variance of the distribution. The first two moments, *a* the intercept and *b* the slope are as follow:

$$a = \overline{SSM_m} - b \times \overline{SSM_o} \tag{1}$$

$$b = \frac{\sigma_m}{\sigma_o} \tag{2}$$

where $\overline{SSM_m}(\sigma_m)$ and $\overline{SSM_o}(\sigma_0)$ correspond to the model and observation means (standard deviations), respectively. The authors of [33,71] discussed the importance of allowing for seasonal variability in the CDF matching. *a* and *b* parameters vary spatially and were derived on a monthly basis by using a three-month moving window after screening for presence of ice and urban areas over (i) January 2008 to October 2018 when using ERA5 as atmospheric forcing and (ii) April 2016 to December 2018 when using IFS HRES as atmospheric forcing.

LAI, defined as one-sided area of green elements of the canopy per unit horizontal ground area is observable from space and quantifies the thickness of the vegetation cover. Several LAI collections/versions are available from the CGLS project from 1999. They are retrieved from the SPOT-VGT (from 1999 to 2014) and then from PROBA-V (from 2014 to present) satellite data according to the methodology proposed by [72]. This study makes use of the GEOV2, 1km spatial resolution and 10-day steps in near real time product. Its development has followed several steps including (1) applications of a neural network for providing instantaneous estimates from SPOT-VGT reflectances, (2) a multi-step filtering approach to eliminate contaminated data (e.g., affected by atmospheric effects and snow cover), and (3) temporal techniques for ensuring consistency and continuity as well as short term

projection of the product dynamics [73] (LAI Product User Manual, https://land.copernicus.eu/global/sites/cgls.vito.be/files/products/GIOGL1_PUM_LAI300m-V1_I1.60.pdf, last access January 2019).

## 2.3. ECMWF Atmospheric Forcing

LDAS-Monde is driven by near-surface meteorological fields from both ECMWFs' reanalysis, ERA5, released in 2018, as well as its high resolution operational high resolution weather analysis and forecasts (HRES). ERA5 underlying model and data assimilation system are very similar to that of the operational weather forecast. ERA5 production cycle (IFS Cycle 41r2) is still close to that of the HRES (IFS Cycle 41r2 to 43r3 from 2016 and 45r1 from June 2018, more information at https://www.ecmwf.int/en/forecasts/documentation-and-support/changes-ecmwf-model, last access January 2019). The main difference between the two is the horizontal resolution with 31 km in ERA5 and 9 km in HRES. Another difference is the data assimilation time window which is from 21:00 UTC to 09:00 UTC in ERA5 and from 21:00 UTC to 03:00 UTC in HRES, as it allows more observations to be assimilated in ERA5. The shorter time window in HRES is due to ECMWF operational constraints to deliver timely forecasts.

The ERA5 forcing data includes the lowest model level (about 10-meters above ground level) air temperature, wind speed, specific humidity and pressure and the downwelling fluxes of shortwave and longwave radiation as well as precipitation partitioned in solid and liquid phases. ERA5 forcing data is extracted from the forecasts initialized daily at 00:00 UTC and 12:00 UTC using the hourly forecasts from +1 to +12 h. HRES forcing data is extracted from the forecasts initialized at 00:00 UTC and 12:00 UTC also using the forecasts from +1 h to 12 h. The same downwelling fluxes as in ERA5 are used but for HRES 2-meter temperature and dewpoint temperature, 10-meters wind-speed are extracted. Specific humidity is then calculated from 2-meter temperature and dewpoint temperature. HRES also has the lowest model level data archived, but due to data storage and access constraints it was more efficient to extract the 2-meter temperatures and 10-meter wind speed. Despite the difference in extracting the near-surface fields, lowest model level and 2-meter temperature and 10-meters winds are very similar, and this is not expected to impact substantially the results. In ERA5 and HRES, the +1 h to +12 h hourly forecasts were concatenated to generate continuous time series. Atmospheric forcing is interpolated from the native grids of ERA5 and HRES to regular grids of $0.25° \times 0.25°$ and $0.1° \times 0.1°$, respectively. The grid to grid interpolation was performed via a bilinear interpolation using the four neighbors in the source grid fitting latitude and longitude linearly. From the forecast initialized at 00:00 UTC, HRES is also available up to 10-days ahead. HRES forecast step frequency is hourly up to time step 90, 3-hourly from time-step 93 to 144 and 6-hourly from time-step 150 to 240 (i.e., 10 days). While the original 3-hourly time steps are used up to day 6 (time step 144), the 6-hourly time steps from day 6 to 10 are interpolated to 3-hourly frequency.

## 2.4. Experimental Setup

Table 1 presents the different experiments evaluated in this study. LDAS-Monde is first forced by ERA5 from 2008 to October 2018 (LDAS-ERA5 hereafter) and HRES (LDAS-HRES hereafter) from April 2016 to December 2018 over a western Europe domain (defined as longitudes from 10.5°W to 20.5°E, latitudes from 42°N to 59°N). IFS is obtained from frequently updated versions of operational system at ECMWF (including changes in spatial and vertical resolutions, data assimilation, parameterizations, and sources of data), while reanalysis like ERA5 guarantees a higher level of consistency (e.g., same model) over long time period because of its frozen configuration. From April 2016 onward, IFS has a spatial resolution of about $0.1° \times 0.1°$ (HRES). Despite the spatial resolution, ERA5 being a recently released dataset, its production cycle (IFS Cycle 41r2) is still close to that of the HRES (IFS Cycle 41r2 to 43r3 from 2016 and 45r1 from June 2018). At the ERA5 spatial resolution, large scale, long time experiments are computationally affordable, and HRES can be used to focus on specific domains or events.

Vegetation outputs from this set of 4 experiments (assimilation of SSM and LAI as well as their model counterpart, i.e. open-loops without assimilation) are then evaluated. Vegetation from another experiment (model only, without assimilation) is evaluated: ISBA forced by ERA5 down-scaled to HRES spatial resolution (from 0.25° × 0.25° to 0.10° × 0.10°) for 1 year (2017). Additionally to the LDAS-HRES analysis experiment, daily forecast experiments with 8-day lead time (from LDAS-HRES analyzed initial conditions) were also performed over 2018. Forecast experiments with 2 days and 8 days lead time (LDAS_fc_2d and LDAS_fc_8d, respectively) are evaluated.

## 3. Results and Discussions

This section describes the results obtained in this study. The impact of the heatwave on LAI and SSM is first assessed using CGLS satellite derived data. To that end, the latter are expressed in monthly anomalies (difference to the mean scaled by the standard deviation). LDAS-Monde ability to represent LAI and SSM is then assessed using the correlation coefficient (R) and root mean square differences (RMSD).

### 3.1. Monitoring the Heatwave Impact on LAI and SSM Using Remote Sensing

Time-series on Figure 1 illustrate monthly anomalies for CGLS products GEOV2 LAI (over 2000-2018, Figure 1a) and ASCAT SSM (over 2008–2018, Figure 1b) averaged over the domain of interest (presented by Figure 2). On both time-series, July is highlighted in red and the horizontal dashed lines represent the value of July 2018. As for LAI (Figure 1a), July 2018 exhibits a large negative anomaly, greater than twice standard deviations (stdv) on average. Such a low value is not observed in this 19-yr time-series for a month of July and only one month, in summer 2003: August 2003 presents an anomaly value below than that of July 2018. In 2003, large parts of Europe were affected by record-breaking temperature in summer (e.g., [74]). June to October 2018 presented negative LAI anomalies. Table 2 presents the fraction of the considered domain affected by negative anomalies greater than 2 stdv for all months of July over 2000–2018 for GEOV2 LAI and 2008–2018 for ASCAT SSM. In July 2018, it represents nearly 19% of the domain for LAI, the largest percentage observed in 19-year. Not only the 2018 summer heatwave lead to very large negative anomaly values in LAI but it has affected a large part of the domain. Figure 2a shows maps of anomaly for July 2018 for GEOV2 LAI.

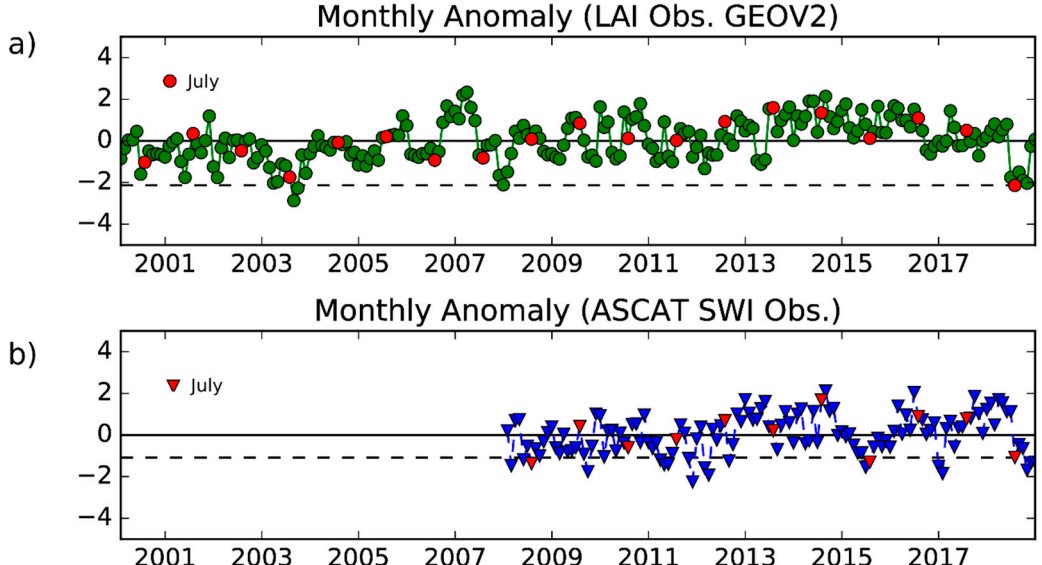

**Figure 1.** Monthly Anomaly time-series (scaled by the standard deviation) of satellite derived (**a**) GEOV2 Leaf Area Index over 2000–2018 and (**b**) Surface Soil Moisture over 2008–2018 from the Copernicus Global Land Service averaged over the domain (presented by Figure 2). Months of July are highlighted in red, straight dashed lines represent values for July 2018.

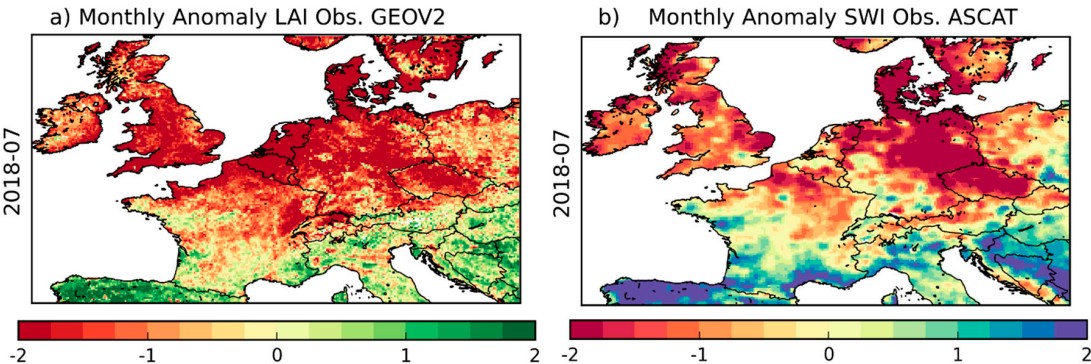

**Figure 2.** Monthly anomalies (scaled by standard deviation, expressed in units of standard deviation) maps for July 2018 for (**a**) GEOV2 Leaf Area Index with respect to 2000–2018 and (**b**) Surface Soil Moisture with respect to 2008–2018 from the Copernicus Global Land Service. Green/red colors represent positive/negative anomalies in units of standard deviation.

**Table 2.** Percentage of the domain with monthly anomalies lower than -2 stdv for satellite derived GEOV2 Leaf Area Index, ASCAT surface soil moisture. Only months of July are represented.

| | July 2000 | July 2001 | July 2002 | July 2003 | July 2004 | July 2005 | July 2006 | July 2007 | July 2008 | July 2009 | July 2010 | July 2011 | July 2012 | July 2013 | July 2014 | July 2015 | July 2016 | July 2017 | July 2018 |
|---|---|---|---|---|---|---|---|---|---|---|---|---|---|---|---|---|---|---|---|
| GEOV2 Leaf Area Index | 5.0 | 0.4 | 0.2 | 5.0 | 0.6 | 0.8 | 1.8 | 1.1 | 0.2 | 0.03 | 0.7 | 0.7 | 0.3 | 0.7 | 0.2 | 2 | 0.1 | 0.6 | 19 |
| ASCAT SWI | N/A | N/A | N/A | N/A | N/A | N/A | N/A | N/A | N/A | 2 | 0.04 | 1.7 | 0.2 | 1.5 | 0.5 | 0.06 | 3.0 | 0.01 | 10 |

Figure 2a shows that most of the UK, Northern France, Belgium, Netherlands, Denmark, Germany and Czech-republic present anomaly values greater than -2 stdv. ASCAT SSM exhibits large negative anomalies for July 2018 (Figure 2b) with values greater than -1 standard deviation. Such low values were also observed in July 2008 and 2015, and it is worth noticing from Table 2 that in July 2018, 10% of the domain was affected by anomalies greater than -2 stdv, while only ~2% and ~3% for July 2008 and 2015. Figure 2b (maps on anomaly for July 2018 for ASCAT SSM) shows that the southern part of the domain presents large positive anomalies values (e.g., north of Spain, in the Balkans) and that

good geographical agreement between GEOV2 LAI and ASCAT SSM anomalies is found. While some winter months show large negative anomalies in ASCAT SSM, e.g., December 2010, 2011, this might be related to frozen conditions not accounted for and interpreted as dry conditions.

### 3.2. Monitoring the Heatwave Impact on Vegetation Using LDAS-Monde

LDAS-Monde, being an offline reanalysis of the land surface variables, is forced by atmospheric datasets: ERA5 and HRES in this study. Using both datasets to force LDAS-Monde produces a long reanalysis of the LSVs (from the use of ERA5) with real-time and even forecast capacity (from the use of HRES). As ERA5 is available over a large temporal extent (from 2000 at the time of study) it can be useful to study recent climate changes. Anomaly time-series of air temperature and precipitation from ERA5 are presented in Figure 3. While it is not our intention to repeat the study from [49] on predicting the summer 2018 heatwave, Figure 3 highlights that the April to August period in 2018 exhibits rather large positive anomaly values of air temperature (Figure 3a) with July 2018 being the highest value observed between January 2001 and October 2018. For precipitation, all months from May to October 2018 present large negative anomalies with July 2018 being the third lowest within the considered period. One may also note the coherence between air temperature and precipitation from ERA5 and the satellite derived observation presented above for this 2018 heatwave event, particularly for LAI. As seen from Figures 1a and 3, large positive anomalies of air temperature (Figure 3a) are associated with large negative anomalies of precipitation (Figure 3b) as well as large negative anomalies of LAI (Figure 1a). In the beginning of 2007 temperature and precipitation show positive anomalies which reflect on LAI presenting large positive anomalies. While in the beginning of 2013, both air temperature and LAI show negative anomalies.

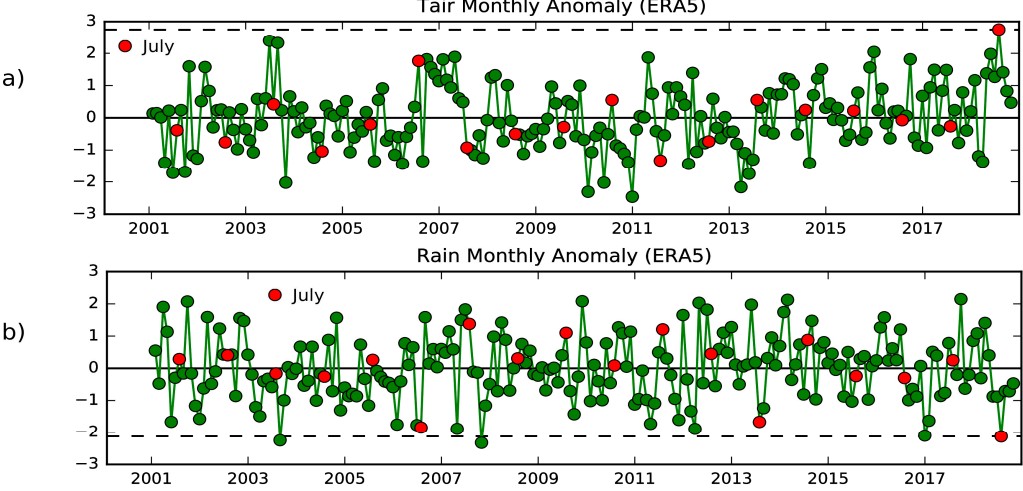

**Figure 3.** Monthly Anomaly time-series (scaled by the standard deviation, expressed in units of standard deviation) of air temperature (**a**) and precipitations (**b**) from ERA5 atmospheric reanalysis dataset over January 2001–October 2018. Months of July are highlighted in red, straight dashed lines represent values for July 2018.

When LDAS-Monde is driven by ERA5 and integrates LAI and SSM through data assimilation, those anomalies should be reflected on analyzed land surface conditions and their impact propagated to other land surface variables through biophysical processes and feedback in the model. Figure 4a illustrates observed CGLS GEOV2 Leaf Area Index (LAI), over 2008–2018 as well as LDAS-Monde LAI time-series forced by either ERA5 (LDAS-ERA5) over January 2008–October 2018 or HRES (LDAS-HRES) over April 2016–December 2018. Figure 4b shows the same as Figure 4a for the common April 2014 to October 2018 period. From Figure 4 one may notice the good agreement between the analyzed LAI and the observed annual cycle. While neither the open-loop nor the analysis capture

the maximum LAI peak well (as already observed by [18]), the analysis efficiently corrects for the open-loop delay during the senescence phase. Considering the period where both ERA5 and HRES are available to force LDAS-Monde (April 2016 to October 2018), one may notice the relative good agreement between LDAS-ERA5 and LDAS-HRES, both in the open-loops and analyses, as well as the senescence phase being very well represented by LDAS-HRES analysis (which failed to capture the LAI peak intensity though).

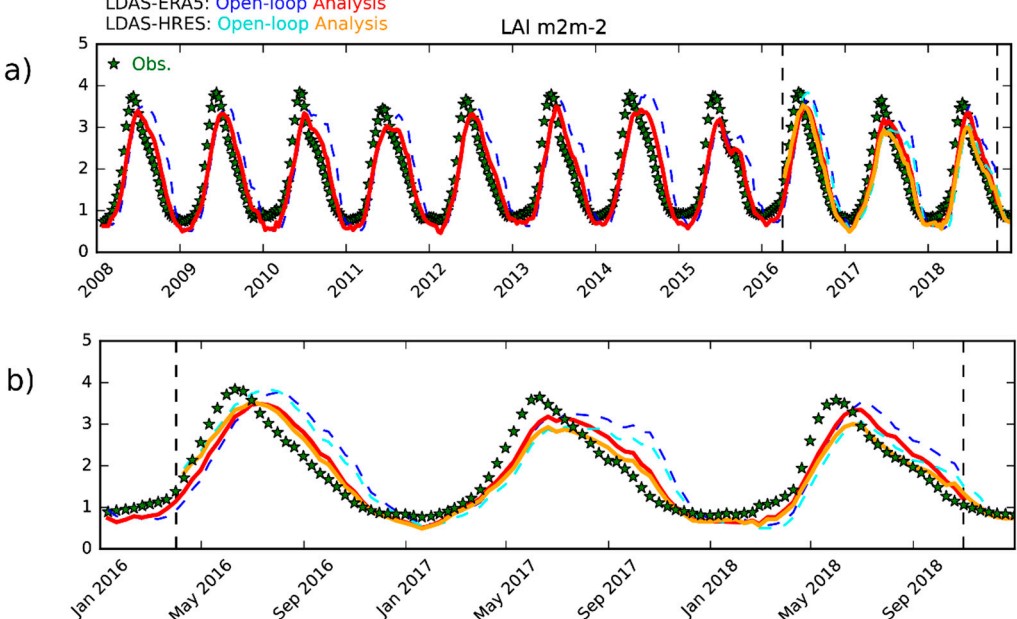

**Figure 4.** (**a**) Observed CGLS GEOV2 Leaf Area Index (LAI) (green stars) over January 2008 to December 2018 as well as LDAS-Monde LAI time-series forced by either ERA5 (Open-loop is in blue, analysis is in red) over January 2008–October 2018 or high resolution operational analysis (HRES) (Open-loop is in cyan, analysis is in orange) over April 2016–December 2018. (**b**) Same as a) over Land Data Assimilation System (LDAS)-HRES and LDAS-ERA5 common period (April 2016 to October 2018). Data are averaged over the domain illustrated by Figure 2, dashed line represents the date from when HRES is available (April 2016) and the date up to when ERA5 is available (at the time of the study).

Upper panel of Figure 5 illustrates seasonal RMSD (Figure 5a) and correlation (Figure 5b) values between LAI from the model forced by either ERA5 (LDAS-ERA5 Open-loop) or HRES (LDAS-HRES Open-loop), the analysis forced by either ERA5 (LDAS-ERA5 Analysis) or HRES (LDAS-HRES Open-loop) and GEOV2 LAI estimates from CGLS from April 2016 to October 2018. Figure 5 lower panel shows the same between modeled/analyzed soil moisture from the second layer of soil (1–4 cm) and ASCAT surface soil moisture estimates from CGLS (and converted into the model space, in $m^3m^{-3}$, as detailed in Section 2.1). From Figure 5 (all panels), one may see that LDAS-ERA5 and LDAS-HRES open-loops are quite comparable, LDAS-HRES open-loop being slightly better than LDAS-ERA5 open-loop in representing both LAI and soil moisture. It is also visible that the analyses add skill to both open-loops for both variables, which indicates the healthy behavior from the land data assimilation system. Over the whole common period (from April 2016 to October 2018), averaged R and RMSD values for LDAS-ERA5 open-loop (analysis) are 0.575(0.798) and 1.215 $m^2m^{-2}$ (0.796 $m^2m^2$) for LAI, 0.748(0.772) and 0.038 $m^3m^{-3}$ (0.035 $m^3m^{-3}$) for soil moisture, respectively. For LDAS-HRES, they are 0.601(0.808) and 1.150 $m^2m^{-2}$ (0.772 $m^2m^{-2}$) for LAI and 0.750(0.772) and 0.038 $m^3m^{-3}$ (0.036 $m^3m^{-3}$), respectively. It is also worth mentioning that if the analysis brings a clear improvement in the representation of LAI, reducing the overestimation in the senescence phase, as its model counterpart, it fails capturing the observed LAI onset. Albergel et al. [7] has evaluated the model sensitivity of the observation for Europe over 2000–2012 reflected in the SEKF Jacobians. The Jacobians depend on the

model physics and their examination provides useful insight that explains the data assimilation system performances [34,65]. Albergel et al. [7] suggests a seasonal dependency of the model sensitivity to the observed LAI. High sensitivity is found in autumn. Smaller model sensitivity at the time of the year where the LAI onset occurs (spring) prevails the analysis to match the observations correctly.

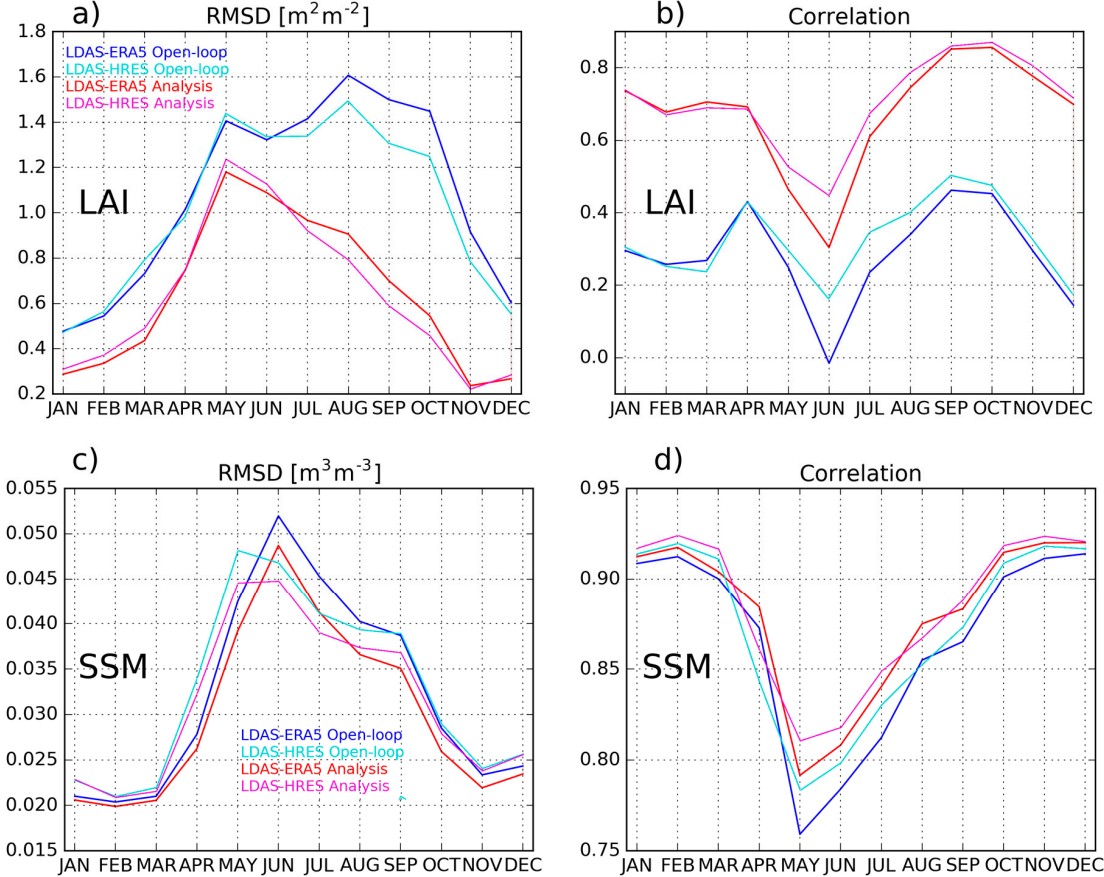

**Figure 5.** Upper panel, seasonal (**a**) root mean square differences (RMSD) and (**b**) correlation values between leaf area index (LAI) from the model forced by either ERA5 (LDAS-ERA5 Open-loop in blue) or HRES (LDAS-HRES Open-loop in cyan), the analysis forced by either ERA5 (LDAS-ERA5 Analysis in red) or HRES (LDAS-HRES Open-loop in pink) and GEOV2 LAI estimates from the Copernicus Global Land Service project from April 2016 to October 2018. Lower panel, same as upper panel between modeled/analyzed soil moisture from the second layer of soil (1–4 cm) and ASCAT surface soil moisture estimates from the Copernicus Global Land Service project.

As highlighted by [75], who have evaluated the capacity of several LSMs (including ISBA) to accurately simulate observed energy and water fluxes during droughts, there is a need to re-examine existing model components in LSMs to improve simulations of soil hydrological processes and water–plant interactions.

Finally, Figure 6 shows LAI for the month of July 2018 from the open-loop, observations, analysis as well as LAI differences (analysis minus open-loop) for LDAS-ERA5 (upper panels, $0.25° \times 0.25°$ spatial resolution) and LDAS-HRES (lower panels, $0.10° \times 0.10°$ spatial resolution). From the two open-loops, one can see that LDAS-ERA5 and LDAS-HRES overestimate LAI with respect to the observations. LDAS-HRES open-loop is however in better agreement with the observations than LDAS-ERA5 open-loop, particularly over the area most affected by the heatwave (e.g., over Belgium, the Netherlands, Germany and Poland). While the assimilation is efficiently reducing LAI in both LDAS-ERA5 and LDAS-HRES analyses, the latter is in better agreement with the observations than LDAS-ERA5 analysis. Despite their spatial resolution differences, ERA-5 and HRES results present

similar LAI patterns. They both underestimate the amplitude and spatial extent of the drought in the open-loop, and for both the analysis effectively improves the particular LAI conditions associated to the 2018 heatwave. Furthermore, due to the large-scale nature of the drought event the spatial resolution differences between ERA5 and HRES do not affect significantly the simulations.

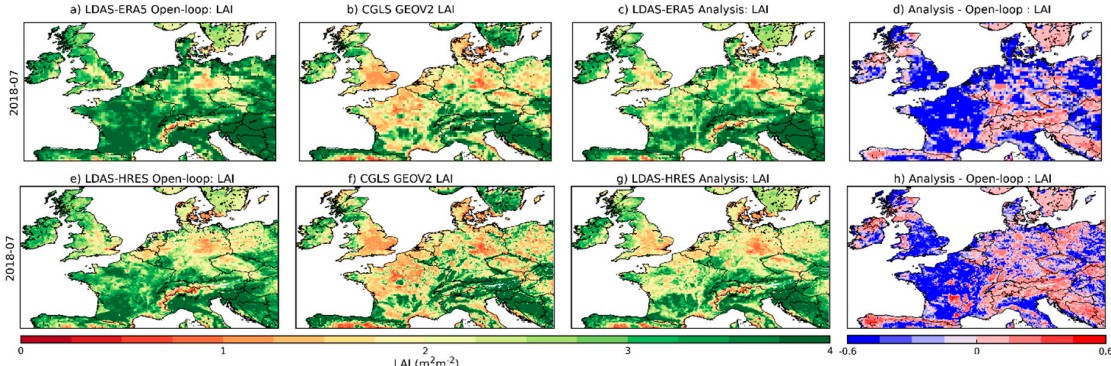

**Figure 6.** Upper panel, Leaf Area Index from (**a**) LDAS-ERA5 Open-loop, (**b**) the observations, (**c**) LDAS-ERA5 Analysis and (**d**) differences between LDAS-ERA5 Analysis and LDAS-ERA5 Open-loop for July 2018. Lower panel, (**e**–**h**) same as upper panel for LDAS-HRES. Spatial resolution of upper panel is $0.25° × 0.25°$, spatial resolution of lower panel is $0.10° × 0.10°$.

Figure 7 represents maps of monthly anomalies from LDAS-ERA5 for July 2008, 2010, 2012, 2014, 2016 and 2018 for soil moisture in the fourth layer of soil (wg4, between 20 cm and 40 cm) as well as drainage, runoff and evapotranspiration over most of the UK. July 2018 presents the strongest negative anomalies. Figure 7 is complementary to Figures 4–6 illustrating how the data assimilation system impacts the model variables. Assimilating SSM and LAI within LDAS-Monde results in updates of the LSM variables in different ways. Control variables in the state vector (LAI and soil moisture from several layers of soil) are first updated through the Kalman gain computed by the SEKF. That gain especially provides sensitivity to observed variables (LAI and the second layer of soil, wg2, between 1 cm and 4 cm) for unobserved variables allowing their updates by the SEKF (e.g., soil moisture from the fourth layer of soil, wg4, top row of Figure 7). Then, other variables are indirectly modified by the analysis through biophysical processes and feedback in the model by updates of the control variables (e.g., drainage, runoff and evapotranspiration as seen on Figure 7). It is worth mentioning the positive anomaly values for July 2012, particularly in runoff and drainage responding to persistent rain during the first weekend of July that had led to flooding in many part of the UK (https://www.metoffice.gov.uk/learning/learn-about-the-weather/weather-phenomena/case-studies/july-2012-flooding, last access January 2019).

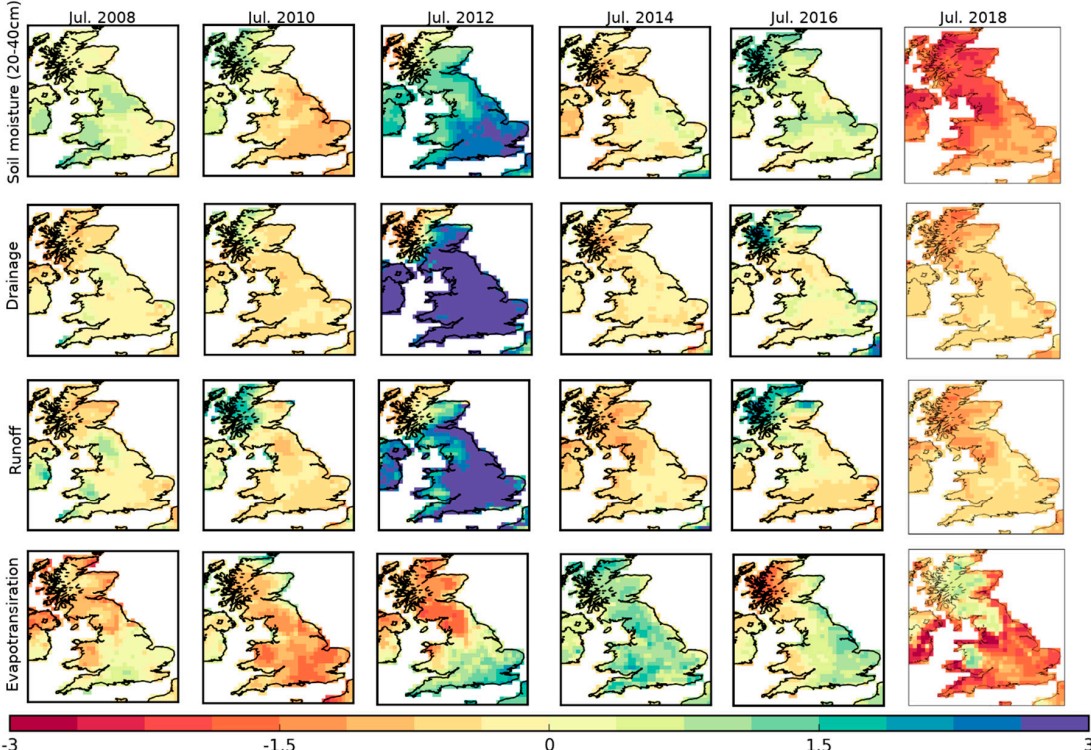

**Figure 7.** Maps of monthly anomalies (expressed in units of standard deviation) from LDAS-ERA5 analysis for July 2008, 2010, 2012, 2014, 2016 and 2018 with respect to the 2008–2018 period (from left to right) for the following variables: soil moisture form the fourth layer of soil (between 20 cm and 40 cm), drainage, runoff and evapotranspiration (from top to bottom).

Both LDAS-Monde configurations forced by either ERA5 or HRES lead to an accurate representation of vegetation during the summer 2018 heatwave and in general. The HRES configuration presents slightly better results over the common period investigated. HRES being obtained from frequently updated versions of the IFS it is not a fixed system in time, while a reanalysis like ERA5 guarantees a higher level of consistency because of its frozen configuration. ERA5 has a coarser spatial resolution than the HRES. Its spatial resolution allows, however LDAS experiments to be long term and affordable at large scale. With ERA5 available back to 1950 and covering near real-time needs with the ERA5T (https://climate.copernicus.eu/climate-reanalysis, last access March 2019), an LDAS-ERA5 would be able to provide a model climate as reference for anomalies of the land surface conditions. Significant anomalies could then be used to trigger more detailed monitoring and forecasting activities for a region of interest using, for example the LDAS-HRES.

The Summer 2018 heatwave clearly had an impact on vegetation and soil moisture, as seen using satellite derived estimates of LAI and SSM. Those satellite estimates are very useful to monitor extreme events impacts but their use is limited by their temporal frequency of a few days at best. While microwave remote sensing provides a way to quantitatively describe the water content of a shallow near-surface soil layer, [76], the variable of interest for applications in short- and medium-range meteorological modeling and hydrological studies over vegetated areas is the root-zone soil moisture content which controls e.g., plant transpiration [68]. Similarly, estimates of above-ground biomass might be more useful than LAI for applications linked to agriculture. Integration of these satellite derived datasets into LSMs through data assimilation is, therefore of paramount importance to improve monitoring accuracy of extreme events impacts on LSVs. Not only the representation of LAI and SSM in such system would be improved but other model variables will benefit from the assimilation through biophysical processes and feedback in the model too [7,10,18,77].

### 3.3. Resolution vs. System Evaluation

Results presented above showed that driving the LDAS by either ERA5 or HRES leads to good results monitoring the impact of the summer 2018 heatwave on vegetation, with HRES providing better results. In an attempt to investigate whether the improvement from the use of ERA5 to HRES is due to the resolution only (e.g., better representation of land cover) or to the forcing quality (or both), another experiment was carried out for 2017 (see Table 1). ERA5 was downscaled from $0.25° \times 0.25°$ to $0.10° \times 0.10°$ (ERA5_010) spatial resolution to force ISBA and outputs were compared to those of LDAS_HRES open-loop (ran for 2017, with similar initial conditions). A bilinear interpolation from the native grid to the regular grid was made. Figure 8 illustrates monthly scores (R and RMSD values over 2017) for LAI from 2 experiments, namely ERA5_010 and LDAS_HRES open-loop. From the two panels of Figure 8, one may appreciate the score similarities between ERA5_010 and LDAS_HRES open-loop. HRES was upscaled to ERA5 spatial resolution to run ISBA and outputs where compared to these of LDAS-ERA5 open-loop (ran for 2017, with similar initial conditions), and similar results as discussed above were obtained (not shown). Although a longer time period would be required to further test such configurations, it is very interesting to notice than when ERA5 forcing is downscaled to $0.10° \times 0.10°$ to force ISBA, it performs almost as good as the operational forcing, HRES. These results could justify running longer periods of time of ERA5 at $0.10° \times 0.10°$ when the operational forcing is not available (e.g., prior to April 2016).

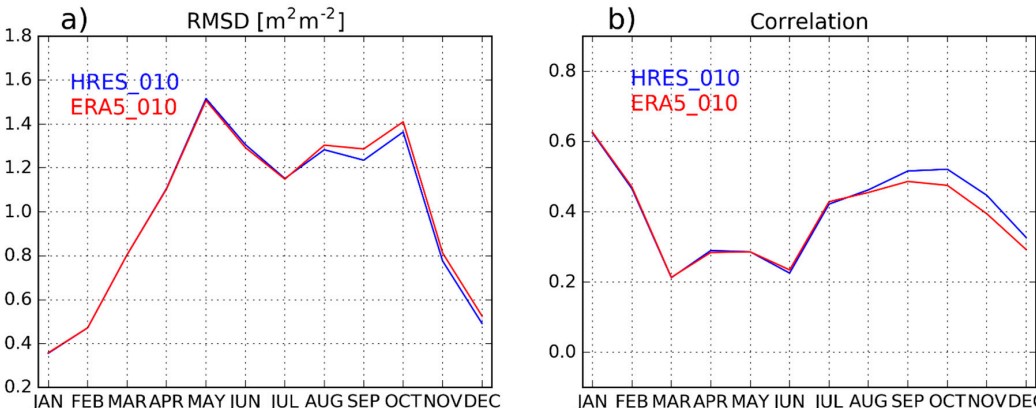

**Figure 8.** Monthly (**a**) RMSD and (**b**) correlation values between leaf area index (LAI) from the model forced by either HRES_010 or ERA5_010 (ERA5 forcing down-scaled to HRES spatial resolution) and GEOV2 LAI estimates from the Copernicus Global Land Service project for the year 2017.

### 3.4. Anticipating the Impact of Heat Waves on Vegetation

Two other experiments are presented in order to (i) study the possibility of forecasting the impact of extreme events on vegetation few days in advance and (ii) highlighting the fact that a forecast initialized by an analyzed state can have more skills than an open loop. For the whole 2018 and for each daily analysis from LDAS-HRES, 2 forecast experiments (2-day and 8-day forecast, see Table 1) were conducted. The atmospheric forcing forecast is coming from HRES, as described in the materials and methods sections. For the sake of clarity, only forecasts with lead time of 2 and 8 days are presented (LDAS_fc_d2 and LDAS_fc_d8, respectively). Figure 9a illustrates LAI time-series from the open-loop, the analysis (ran for 2018, only) as well as the 2 forecast experiments (LDAS_fc_d2 and LDAS_fc_d8) for 2018 averaged over a domain defined as: longitudes from $4°W$ to $15°E$ and latitudes from $48°N$ to $55°N$. According to Figure 2, this domain was more severely affected by the heatwave, and is represented by Figure 9c. Firstly, the large error between all the experiments and the observations for the start of the growing season is noticeable (Figure 9a). Secondly, from March to June LDAS_HRES analysis as well as LDAS_fc_d2 and LDAS_fc_d8 are only slightly correcting for this issue. This is a known issue as already mentioned by [18], the $CO_2$-responsive version of ISBA is such that during the growing phase, enhanced photosynthesis corresponds to a $CO_2$ uptake, which results in vegetation

growth from a prescribed LAI minimum threshold (1 m$^2$m$^{-2}$ for coniferous forest or 0.3 m$^2$m$^{-2}$ for other vegetation types). These thresholds are likely too low and are currently being revisited using the CGLS LAI long term dataset. This is expected to lead to better representation of LAI during the vegetation growing phase [78]. However, during the senescence phase (see zoom on Figure 9b), the analysis is quite efficient in reducing the differences with the observed LAI and it is interesting to notice that so are the 2-d and 8-d forecasts of LAI initialized by the analysis. This suggests that the impact of assimilating satellite observations in LDAS-Monde has the capacity to mitigate model deficiencies, leading to better estimates of the system states and that this impact can last in time. From the two first panels of Figure 9, one may see that LDAS_fc_d2 and LDAS_fc_d8 are closer to the observations than the open-loop. Figure 9c represents RMSD values between the open-loop (ran for 2018, only) and the LAI GEOV2 observations and Figure 9d the RMSD differences between the open-loop (analysis) and the LAI GEOV2. Negative (blue shaded areas) values indicate areas where the analysis has smaller (i.e., better) RMSD values than the open-loop. Figure 9d is dominated by negative values with blue shaded areas representing 82% of the domain. It shows the added value of the analysis over the open-loop. Finally, Figure 9e presents RMSD differences between the open-loop (LDAS_fc_d8) and the LAI GEOV2 observations and it is very interesting to notice than an 8-day forecast initialized by an analysis presents better skills in capturing LAI than an open-loop for most of the domain with blue shaded areas (negative values) still representing ~49% of the studied domain.

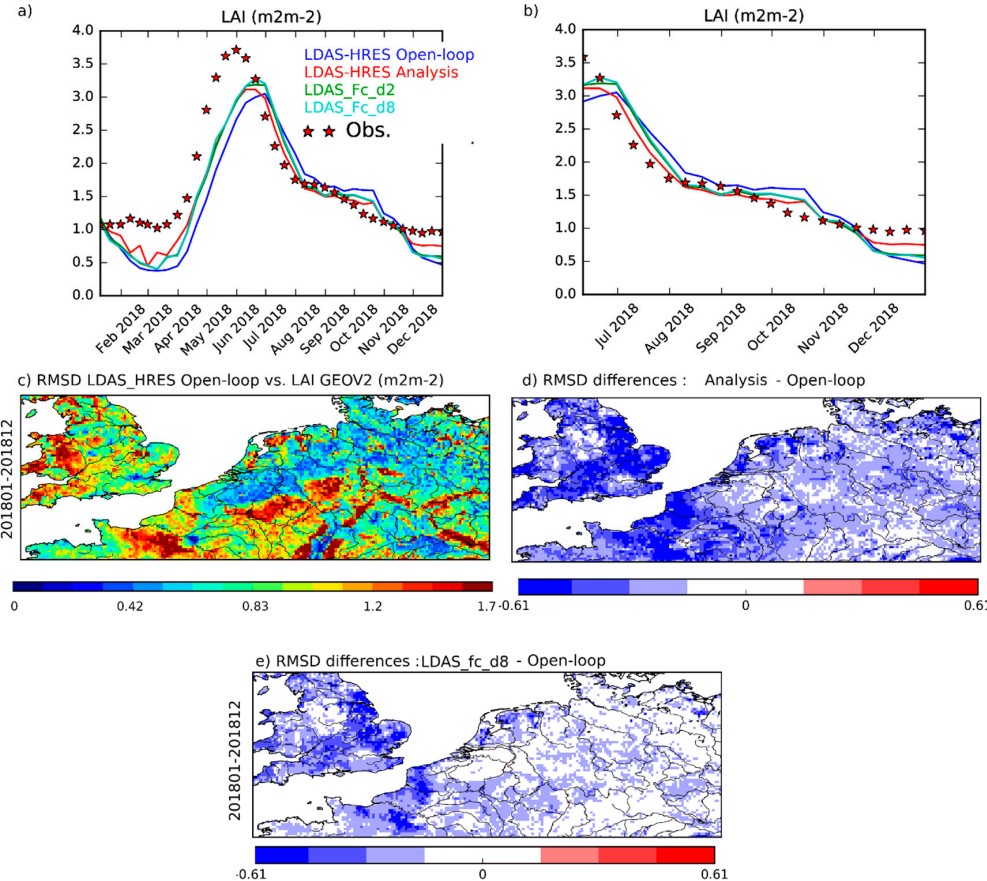

**Figure 9.** (**a**) LAI time series from the model (LDAS-HRES Open-loop in blue), the analysis (LDAS-HRES Analysis in red), the 2-d and 8-d forecasts from the analysis (LDAS_Fc_d2 in green, LDAS_Fc_d8 in cyan respectively) as well as the observations from the Copernicus Global Land Service (LAI GEOV2, red stars) for 2018. (**b**) same as (**a**) focusing on the June–December period. (**c**) RMSD values between LDAS-HRES Open-loop ran over 2018 and LAI GEOV2, (**d**) RMSD differences between LDAS-HRES analysis (open-loop) and LAI GEOV2, (**e**) same as (**d**) for LDAS_fc_d8 and open-loop.

This result is emphasized by Figure 10 showing monthly RMSD and R values between LAI from the four above-mentioned experiments (LDAS-HRES open-loop and analysis, LDAS_fc_d2 and LDAS_fc_d8) and the GEOV2 observations over 2018. The RMSD and R values from LDAS_fc_d2 and LDAS_fc_d8 experiments are better than from the open-loop, all year long. They are closer to those from LDAS-HRES analysis than from its open-loop counterpart. As seen on Figure 10b, it is from July 2018 that the differences between the open-loop and the analysis are the strongest. Impact of assimilating LAI and SSM estimates has a time persistence of at least 8 days on LAI. Future work could focus on giving more statistical strength to those results in particular by considering a longer time period as well as looking at other LSVs.

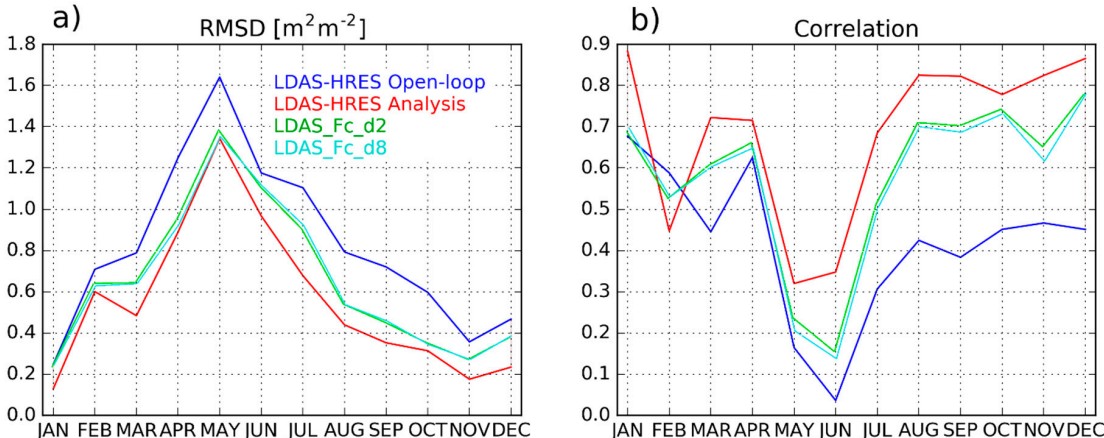

**Figure 10.** Monthly (**a**) RMSD and (**b**) R values between LAI from the model (LDAS-HRES open-loop in blue), analysis (LDAS-HRES Analysis in red), the 2-d and 8-d forecast experiments initialized by the analysis and the Copernicus Global Land Service LAI GEOV2 over 2018.

## 4. Conclusions and Perspectives

This study has investigated the capability of LDAS-Monde offline land data assimilation system to represent the impact of the summer 2018 heatwave on vegetation. Satellite derived leaf area index and surface soil moisture were assimilated in LDAS-Monde forced by either ERA5 reanalyses ($0.25°$ $\times 0.25°$ spatial resolution) or IFS HRES operational product ($0.10° \times 0.10°$ spatial resolution) from ECMWF. Both analysis experiments were able to represent the impact of the heatwave on vegetation well. While there is a surface physiography and modeling advantage to the HRES configuration, there is added value in down-scaling ERA5 to HRES spatial resolution, too. It would allow consistent, long term and high-resolution reanalysis of the LSVs. The possibility of forecasting LSVs was tested and it was shown that a forecast of LAI from analyzed initial conditions has more skill than an open-loop (with a persistence of at least 8 days). Combining ERA5 atmospheric re-analysis, HRES analysis and its forecast within LDAS-Monde is highly relevant to foster research for land applications at various timescales from daily to annual. The use of HRES data to force LDAS-Monde is very promising and it can be complemented by ECMWF 51-member ensemble forecasts (~18 km spatial resolution). Use of the ECMWF ensemble in LDAS-Monde could help to capture uncertainties in the representation of LSVs. It would open the possibility to anticipate the impact of heatwaves at monthly temporal scales using a probabilistic method. Moreover, since one member of the ensemble is similar to HRES at a coarser spatial resolution, and the ensemble is available up to 15-days lead time (twice a day and up to 45 days twice a week) it can be used to test longer range forecast of LSVs than when using HRES.

One of the limitations of the use of the discussed land data assimilation system at a high spatial resolution, for example using grid cells of 1 km or 300 m, is that analyzed atmospheric forcings are not available at these scales. While downscaling atmospheric forcing like the IFS HRES (e.g., from $0.1° \times 0.1°$ to $0.01° \times 0.01°$ spatial resolution) is likely to add uncertainties, their impact on the representation of the LSVs can be reduced through the dynamic integration of satellite-derived LAI

observations at fine scale like the 300 m spatial resolution product from Copernicus Global Land Service. For the meteorological forcing, the use of AROME (Application de la Recherche à l'Opérationnel à Méso-Échelle), the operational numerical prediction model from Météo-France, atmospheric variables to drive the LDAS could also be investigated as its spatial resolution is already of 1.3 km × 1.3 km over France. The process of comparing Land Surface Models and observations, e.g. through data assimilation, can highlight model deficiencies. It is likely that the model would benefit from new LAI minimal values parameterization that are currently being revisited at Météo-France using the long-term CGLS data-set including more than 19-yr of LAI data.

**Author Contributions:** Conceptualization, C.A.; Investigation, C.A.; Methodology, C.A., E.D. and G.B.; Writing—original draft, C.A.; Writing—review and editing, C.A., E.D., B.B., Y.Z., S.M., G.B., P.d.R., J.M.-S. and J.-C.C.

**Funding:** This resarch received no external funding.

**Acknowledgments:** Results were generated using the Copernicus Climate Change Service Information, 2017. The Authors would like to thanks the Copernicus Global Land Service for providing the satellite derived Leaf Area Index and Surface Soil Moisture.

**Conflicts of Interest:** The authors declare no conflict of interest.

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
