# Peer review of "Monitoring and Forecasting the Impact of the 2018 Summer Heatwave on Vegetation"

_remotesensing, doi:10.3390/rs11050520_

Round 1
Reviewer 1 Report
Monitoring and Forecasting the Impact of the 2018 Summer Heatwave on Vegetation
C. Albergel et al.
This study is very interesting and promising. However, the paper does not do it justice. I feel like it was written quickly and that the authors skimmed over some key explanations. A more in-depth explanation of the methodology and analysis of the results are necessary. A thorough read-through is also necessary to correct typos, phrases that are not well constructed and add missing information in tables and figures.
Major comments:
Section 2.2 needs to be developed as the reader needs more explanation to fully understand why choices were made and what are the implications of those choices. For example, why is SWI-001 used as a proxy for SSM (L176), and what does it imply? L179, what about forested areas? L182-185, more information is needed on the method used (why this method, equations, etc…).
L217 – What process is implied in “we processed 2-m temperature”
L295 – Is 18 years enough to see a climate signal?
L322-336 – Define RMSD and R (acronym, equation, units, …). There is a lot more information that can be extracted from figure 5. For example, explain why the LAI shows a dip in correlation in June and why it is shifted in time when compared to SSM.
L351-360 – Same comment. An in-depth analysis of figure 7 is lacking, more discussion is needed.
L398 – What do the authors mean by bilinear interpolation?
L402-403 – Please explain why.
Fig8 – Please explain the differences seen in fall.
Section 4.2 – This section should not be in the discussion but rather in the results section.
Minor comments:
L12 – Remove “the” before “ECMWF”
L16 – Replace “Analysis” by “The study”
L19 – Replace “considered domain” by “domain of interest” or “study domain”
L19 – Remove “the” before “LDAS-Monde”
L21 – same as above
L40-89 – The paragraph is too long, please separate into 2-3 paragraphs.
L45 – Replace “Model (LSM)’s” by “Models (LSM)”
L49 – Remove “the” before LSM
L51 – Replace “lead towards improved” by “improve”
L127 – Define the acronym ISBA
L129 – remove “-“ after “CNRM”
L131-132 – Remove “the” before “various” and “LSVs”
L137 – “later” needs to be changed to “latter” throughout the manuscript
L138 – Replace “of the ISBA LSM” by “in ISBA”
L150 – Remove all “, also.” throughout the manuscript
L159 – Replace “thanks to” with via or through
Table1 – LDAS-ERA5: add month to 2008 in “2008-10/2018”
LDAS-HRES: correct date in “04/2016-20182018”
Separation bar missing in column “Domain & spatial resolution”
L181-182 – Remove “the” in front of “field” and “wilting”
L187 – Remove “practically”
L211 – Replace “height” with “above ground level”
L217 – Please avoid the use of “we”
L226 – Replace “10-d” by 10 days
L258-259 – Please rephrase
L267 – Replace “From fig.2a, it is visible” with “Figure 2a shows”
L267 – Remove “part of”
L269 – Replace “, greater than -1, also” with “with values greater than -1.”
L272 – Please refrain from using “it is visible”
L273 – presents
L274 – Remove “the” in front of “good.
L274-276 – Please rephrase and correct anomaly to anomalies
Table2 – Please stick to the use of one or two decimals throughout the table
L291 – Please correct to “LDAS-Monde, being …, is forced”
L320-321 – Please rephrase to form a full sentence and avoid the use of remarkable which is a little too strong in this case.
L334-335 – add a space before parentheses
Fig3 – What do the red dots correspond to?
Fig5 – Include the name of the variables (LAI and SSM) in the figure.
L402 – The latter performs only slightly better
L405 – remove “also”
L428 – Replace those with these
L435 – Replace these by such
L437-438 – Replace will by would (since it is hypothetical)
L460-461 – refrain from using “quite” twice in the same sentence
L475 – four
L499 – Remove “the” in front of “IFS”
L501 – “advantage to” instead of “advantage of”
L503 – Please rephrase “The possibility of forecasting LSVs was successfully implemented”
L504 – it was shown that and LAI forecast from analyzed…
L505 – skill
L509 – Move the two sentenced of L512-514 and put them before “Moreover”.
L509 – Moreover, since a member of … and the ensemble…, it could be used…
L522 – Méso-Échelle), the operational … Météo-France,
Author Response
We thank anonymous Reviewer#1 for his/her review of the manuscript and for highlighting the relevance of the study. Reviewer#1 has made several fruitful comments/corrections/suggestions that led to an improved version of the manuscript. Again we would like to thank Reviewer#1 for his/her work and his/her help in particular with respect to the editorial comments. Please find attached our responses his/her comments

Reviewer 2 Report
The paper intends to discuss the potential of the LDAS-Monde platform to monitor the impact on vegetation state of the 2018 summer heatwave over western Europe. But the paper is not well organized and presented. The experiments in this paper is described in Table 1. However, the paper text and figures use LDAS-ERA5 “open-loop” and “Analysis”, LDAS-HRES “open-loop” and “analysis”, which cannot be found in Table 1 and it is hard for readers to make their guess because they are too crucial for this paper’s major discussion.
The 2018 heat wave impact on LAI and SWI have been demonstrated in satellite products (Figure 1). The LDAS supposes should reproduce the features in the satellite data. If the LDAS only reproduces the satellite products but with more or less errors, it is not that useful. Direct spatial and temporal interpolation of satellite data should serve the purpose. The paper spends great deal of efforts to discuss the errors in LDAS products. This aspect is probably interesting to the data assimilation persons to work on the assimilation methodology. For general readers, they want to see whether the data assimilation system provide new useful information beyond the original data.
I think the authors should reorganize the paper in a better way with a clear focus. If the paper intends to present the impact of different assimilation methodology, they should provide more information for their model setting. LAI is a key variable. How the model to produce the LAI is crucial but it is missing. The paper only presents the carbon not LAI. From the modeling point of view, there are huge difference between these two although LAI is from carbon. From carbon to LAI consists a lot of processes.
Author Response
p { margin-bottom: 0.25cm; line-height: 120%; }a:link { }We
thank anonymous Reviewer#2for his/her review of the manuscript and for highlighting the
relevance of the study. Reviewer#2has made several fruitful comments that led to an improved version of
the manuscript. Again we would like to thank Reviewer#2for his/her work. Please see attached our responses to his/her comments

Reviewer 3 Report
General comments
This is a comprehensive paper by world-leading scientists on the use of a world-leading land data assimilation system to study an important event, both from a scientific and a societal perspective, namely, the summer heatwave experienced in Western Europe during 2018. The results will be of interest to the scientific community and, likely, other stakeholders. I recommend publication of the article once the authors address the specific comments below.
Specific comments
P. 1
L. 29: enhance -> enhances.
L. 35: Perhaps the authors could quantify this «added value»?
L. 40-43: Perhaps include a few references to back these statements.
P. 2
L. 52: “…as well as the carbon…”.
L. 72: A suggestion: do not start a sentence with an acronym.
L. 75: I think a right parenthesis is missing.
L. 76: Should be “Centre” and “Medium-Range Forecasts”.
L. 77: I suggest: “[42]; …”.
L. 92: I suggest: “…that led to…”.
P. 3
L. 100: Presumably the +2.16C is above climatology. If so, specify.
L. 102: Perhaps the authors could quantify how much weaker was the precipitation.
L. 140: Perhaps the authors could explain what C3 and C4 crops are.
L. 150: Remove “, also.” Omit needless words.
L. 155: I think it would be better to replace “more” with “most”.
L. 156: If you use UK English, it should be “analysed”. Similarly, in L. 157 it should be “metre”.
P. 5
L. 162-163: In Table I, check timeline for experiment LDAS HRES. I do not understand the timeline.
P. 6
L. 179: “The SSM product…”.
L. 186: You define LAI here, but have introduced it earlier.
L. 202, 203: “The ERA5…”. I think you should introduce the article in these cases.
P. 7
L. 217: I think it is better to write “2-meter”. Same elsewhere in the text.
L. 226: “10-days ahead…”.
L. 243: I suggest replace “4” with “four”. I would suggest using this style for numbers up to ten.
P. 8
L. 269: Presumably, “-1 stdv”. If so, specify.
L. 273: present -> presents.
L. 278: I suggest the authors provide more information on the legend in the caption. This would include information on the green circles, specifying that the dashed lines are straight, and identifying the solid straight line. Same for other similar figures, if appropriate.
P. 9
L. 283: I suggest the authors indicate in the caption what the extremes in the colour bars indicate. For example, green/red represent positive/negative anomalies in stdv. Same for other similar figures, if appropriate.
P. 11
L. 297: Why is this interesting? Avoid subjective comments. Same for L. 406, 461, 471.
L. 320: Why is this remarkable?
L. 321: capturing -> to capture.
L. 322: You have not introduced the acronym RMSD.
L. 333: Make sure you introduce the correlation, R.
P. 12
L. 345: Remove “also”. Omit needless words here and elsewhere in the text. In particular, consider whether you need to use “also”, and “too”.
P. 15
L. 392: lead -> leads.
L. 402: later -> latter.
P. 16
L. 420: “…allows, however, …”.
L. 422: Have you introduced ERA5T?
L. 434: applications.
L. 435: “… is, therefore, …”.
P. 17
L. 451: Is there a “secondly” later? If not, avoid the use of “firstly”.
L. 457: probably -> likely. Unless you can relate your statement to a probability distribution.
L. 470: “Finally, …”.
L. 479-480: Make more specific the link between the results and the time persistence. Same for L. 505.
P. 19
L. 504: showed -> shown.
L. 512: capturing -> capture.
L. 515: to -> of.
L. 525: I suggest: “…assimilation, can highlight model…”. Remove “also”.
Author Response
p { margin-bottom: 0.25cm; line-height: 120%; }We
thank anonymous Reviewer#3for his/her review of the manuscript and for highlighting the
relevance of the study. Reviewer#3has made several fruitful comments that led to an improved version of
the manuscript. Again we would like to thank Reviewer#3for his/her work, in particular in
respect to all the editorial comments. Please see attached our responses to his/her comments
